# Web-based support for self-management strategies versus usual care for people with COPD in primary healthcare: a protocol for a randomised, 12-month, parallel-group pragmatic trial

Tobias Stenlund ![ORCID] , André Nyberg, Sara Lundell, Karin Wadell

Department of Community Medicine and Rehabilitation, Physiotherapy, Umeå University, Umeå, Sweden

**Correspondence to**
Dr Tobias Stenlund;
tobias.stenlund@umu.se

## ABSTRACT

**Introduction** The use of adequate self-management strategies for people with chronic obstructive pulmonary disease (COPD) may increase the level of physical activity (PA), improve health-related quality of life (HRQoL) and reduce healthcare use. Whether web-based support in addition to prompts (email and SMS) could be used to promote self-management strategies to facilitate behaviour change in people with COPD is not clear. This clinical trial aims to generate evidence on the effect of a web-based solution, the COPD Web, in a cohort of people with COPD in a primary healthcare context.

**Methods and analysis** The overall design is a pragmatic randomised controlled trial with preassessments and postassessments (3 and 12 months) and an implementation and user experience evaluation. People with a diagnosis of COPD, treated in primary healthcare will be eligible for the study. A total of 144 participants will be enrolled by healthcare professionals at included primary healthcare units and, after fulfilled baseline assessments, randomised to either control or intervention group. All participants will receive usual care, a pedometer and a leaflet about the importance of PA. Participants in the intervention will, in addition, get access to the COPD Web, an interactive self-managed website that aims to support people with COPD in self-management strategies. They will also continuously get support from prompts with a focus on behaviour change.

The effect on participants' PA, dyspnoea, COPD-related symptoms, HRQoL and health economics will be assessed using accelerometer and questionnaires. To identify enablers and barriers for the use of web-based support to change behaviour, semistructured interviews will be conducted in a subgroup of participants at the 3 months follow-up.

**Ethics and dissemination** Ethical approval has been received from the Regional Ethical Review Board in Umeå, Sweden. Dnr 2018-274-31. Findings will be presented at conferences, submitted for publication in peer-reviewed journals and presented to the involved healthcare professionals, participants and patient organisations.

**Trial registration number** NCT03746873

## Strengths and limitations of this study

► Physical activity level will be objectively measured and bring the field forward regarding knowledge about both short-term and long-term effects of using web-based support.

► Information on how and how much the participants have used the chronic obstructive pulmonary disease (COPD) Web will automatically be collected and analysed throughout the full intervention period, which will increase the understanding of the link between the use of the COPD Web and the possible effects.

► The pragmatic design with generous inclusion criteria and many recruiting primary healthcare units could enhance external validity.

► Prompts will be sent continuously as a reminder and strategy to encourage greater exposures to the COPD Web.

► One limitation is that the sample size is large enough for analysing the effect on physical activity level but may not be large enough for all secondary outcomes.

## INTRODUCTION
### Background and rationale

Chronic obstructive pulmonary disease (COPD) is a chronic and disabling disease with substantial morbidity and mortality. The disease has a steady increase in prevalence and is now the third leading cause of death worldwide.[1] The high prevalence places a considerable burden on the healthcare system with a total yearly cost of COPD in Sweden estimated to 13.9 billion SEK.[2] The mean annual total costs for each person with COPD is 67% higher compared with a person without COPD.[3]

The symptom burden of the disease, respiratory symptoms as progressive dyspnoea, fatigue, impaired physical performance, decreased level of physical activity (PA) and health-related quality of life (HRQoL)[4] is a

consequence of the underlying condition and depend on the individuals' adaptation to the illness and their ability to manage their disease.[5 6] Self-management strategies, including strategies to promote change in health behaviour by increasing the individual's knowledge and skills and their confidence in successfully managing their disease, are therefore now an essential part of COPD management.[5] This have shown to reduce breathlessness and impact of COPD in daily life, increase physical performance, level of PA, HRQoL, adherence to medication as well as improve time to recovery after acute exacerbations and reduce overall health-related costs.[5 7 8] An increased level of PA is of utmost importance and something to promote[9] since PA has been shown to be decreased early in the disease progression[10] and degree of PA is considered the strongest predictor of all-cause mortality in people with COPD.[11 12]

Despite that treatment guidelines and literature strongly support that non-pharmacological treatment (ie, education, self-management strategies, exercise training)[13] should be provided, the vast majority of people with COPD are still excluded from these activities.[14 15] Web-based solutions are promising means of delivering health service and may increase level of PA[16 17] as well as reduced use of health services.[18] However, studies evaluating whether web-based support could be used to promote self-management strategies to support increased PA in people with COPD are contradictory. One showed no effect on PA while other studies showed improved PA[19–21] but that the improvement may not be sustained over a long duration.[21]

The COPD Web is a web-based solution, developed by our research group in cocreation with people with COPD, their relatives, healthcare professionals in primary healthcare (PHC) and researchers.[22] In a pilot study on 83 people with COPD,[23 24] promising results with an increased self-reported level of PA were shown. To know whether this is true also for a larger COPD population, an adequately powered randomised controlled trial with short-term and long-term evaluation is needed.

## Objectives

The main aim is to generate evidence on the effect of the COPD Web in a cohort of people with COPD, currently enrolled for usual care within the PHC context in Sweden. This is of importance, as the vast majority of people with COPD are treated within PHC.[13 15] The specific aims are to evaluate the short-term and long-term effect of the use of the COPD Web in an adequately powered group of people with COPD in PHC context, regarding (i) level of PA, (ii) dyspnoea, (iii) HRQoL, (iv) COPD-related symptoms, (v) health economics in relation to healthcare use and (vi) to identify enablers and barriers for the use of web-based support with the COPD Web in order to change behaviour.

We hypothesise that access and use of the COPD Web, in comparison to usual care, will: (i) increase level of objectively measured PA in people with COPD, (ii) decrease dyspnoea, (iii) increase disease-specific HRQoL, (iv) decrease the number of and/or severity of COPD-related symptoms and (v) decrease the number of COPD-related healthcare contacts in PHC.

## Methods and analysis
### Trial design

The design is a pragmatic randomised controlled trial with preassessments and postassessments (3 and 12 months) in addition to user experience and implementation evaluation. The user experience and implementation evaluation is a necessary complement to understand more about enablers and barriers for behaviour change using web-based support. The study is designed as a pragmatic trial[25] meaning that healthcare professionals, primarily COPD nurses, are involved in recruiting participants, the access to the intervention (COPD Web) is given by the researchers, but the intervention itself only uses self-instructional material and prompts (SMS and email). This design aims to minimise the effort from healthcare professionals and increase the possibility of self-management for people with COPD to improve the applicability of the findings to other healthcare settings. The protocol complies with the SPIRIT (Standard Protocol Items: Recommendations for Interventional Trials) recommendations for protocol reporting[26 27] (online supplementary file 1) and the study will be reported according to CONSORT (Consolidated Standards of Reporting Trials) guidelines for pragmatic trials[25] and eHealth.[28]

### Patient and public involvement (PPI)

We did not directly include PPI in this study, but our research group in cocreation with PPI developed the COPD Web used in the study.

### Participants and intervention
#### Study settings

PHC units from different County Councils in Sweden will constitute the study sites. The number of units is not limited; consequently, more units may be included during the study. At present, 25 units are included, 13 of them situated in urban areas and 12 located in smaller cities or rural areas. The number of enrolled citizens at the included units range between 5700 and 20 300 citizens. One unit has no enrolled citizens but acts as a rehabilitation unit that treats patients with a referral from other PHC units. We will include both publicly funded PHC units and private alternatives.

#### Eligibility criteria

The trial will be conducted from 15 November 2018 until 144 participants are included. All people with a diagnosis of COPD (ICD-10:J44:9) who visit involved PHCCs due to their COPD will be eligible for inclusion in the study if they (1) can read and understand Swedish, (2) have a smartphone, tablet or computer with access to internet, (3) do not have dementia or other psychiatric condition that can prevent understanding of the intervention, (4) do not have severe comorbidity that can be considered

as the contributing factor for limitation in PA and (5) do not already use the COPD Web. In the case of exacerbation, the participant has to wait 6 weeks from the start of pharmacological treatment, before being eligible to the study.

## Participant timeline

The recruitment begins at included PHC units. To facilitate the recruitment of participants, the number of included units will not be restricted to nor the units size, location, how they are funded or the type of care and rehabilitation that the unit offers. Written consent from the operational manager has to be fulfilled before recruitment can start.

To increase the possibility of recruiting participants, the number of exclusion criteria are kept to a minimum. The recruitment will take place during the participant's regular visits to the PHC unit where healthcare professionals will give information about the study. People with

COPD interested in participation will have their contact information and results from latest pulmonary function test (if older than 6 months, a new pulmonary function test will be performed) sent to the research group as displayed in table 1. A researcher (TS) will, after verbal agreement, send informed consent form, questionnaires and activity monitor for baseline assessment to the participants' homes. When the written informed consent and the baseline assessment is fulfilled, the participants' are included and randomised to either the control or intervention group. Follow-up measurements with questionnaires and activity monitor will be conducted at 3 and 12 months after inclusion. A semistructured interview will be done after the 3 months follow-up among a convenient sample of the intervention group.

The participants will be contacted by phone before every assessment to ensure a suitable date for the activity monitoring. In case of non-response after any evaluation,

**Table 1** Participant timeline for enrolment, the intervention and assessments

| Timepoint | $t^{-1}$ screening/ consent | $t^0$ baseline | $t^1$ start | $t^2$ 3 months | $t^3$ (interviews) | $t^4$ 12 months |
|---|---|---|---|---|---|---|
| Enrolment | | | | | | |
| Eligibility screen | x | | | | | |
| Informed consent | | x | | | | |
| Allocation | | | x | | | |
| Intervention | | | | | | |
| The COPD Web | | | ⟶ | | | |
| Assessments | | | | | | |
| Sociodemographic (age, sex, anthropometry, diagnosis)* | | x | | x | | x |
| Pulmonary function† | x | | | | | |
| COPD-related symptoms* | | x | | x | | x |
| Dyspnoea* | | x | | x | | x |
| HRQoL* | | x | | x | | x |
| Time spent in physical activity and training* | | x | | x | | x |
| Time being sedentary* | | x | | x | | x |
| Physical activity level (accelerometer)* | | x | | x | | x |
| Implementation*‡ | | | x | x | x | x |
| Response to and interaction with the COPD Web* | | | | x | x | x |
| COPD-related healthcare contacts* | | | | x | | x |
| Enablers and barriers for the use of a web-based solution* | | | | | x | |

Data collection from
*People with COPD.
†Medical records.
‡Statistics from the website.
COPD, chronic obstructive pulmonary disease; HRQoL, health-related quality of life.

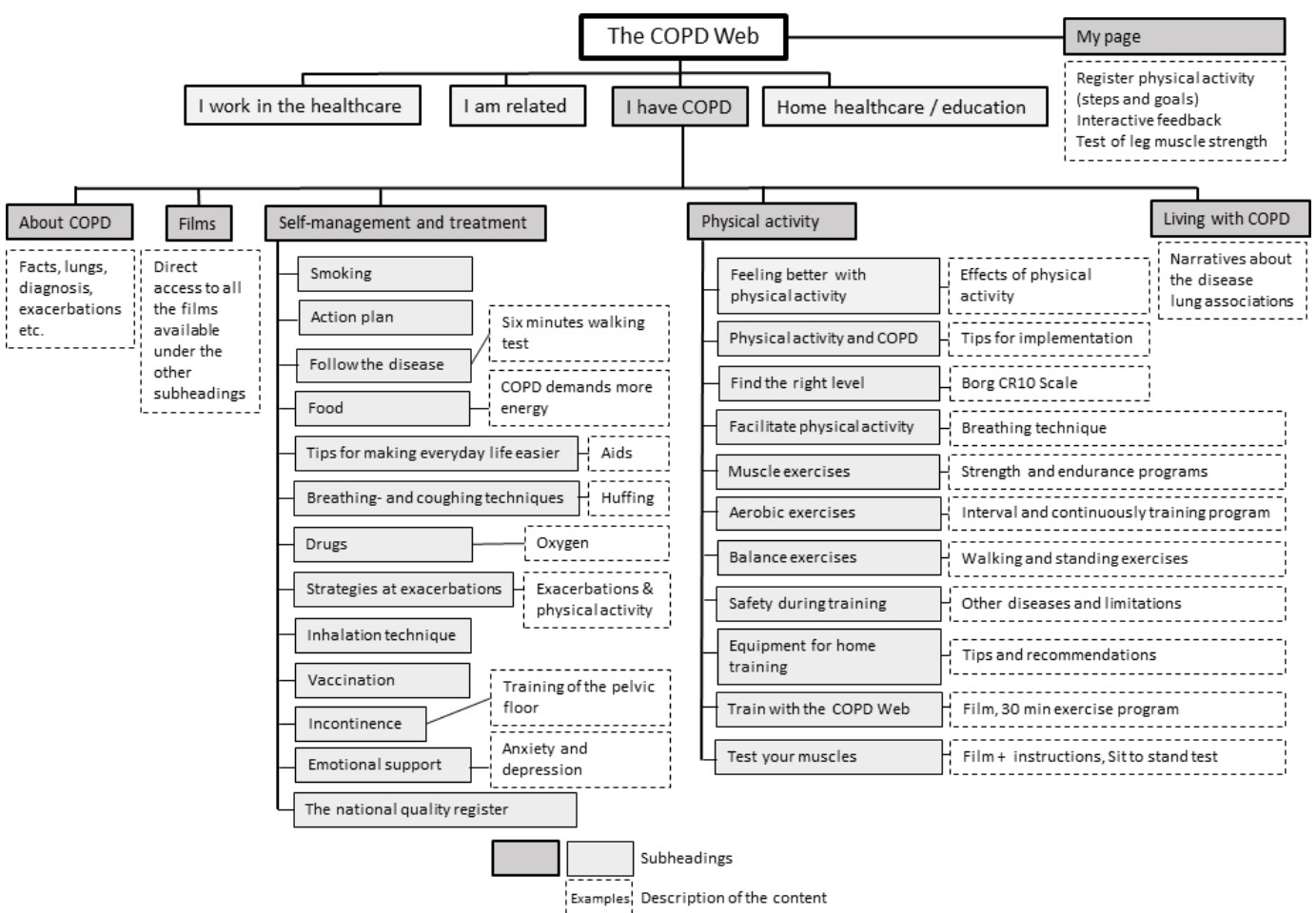

**Figure 1** Figure 1A website map of the COPD Web showing the section 'I have COPD'. COPD, chronic obstructive pulmonary disease.

the participant will be reminded by phone or/and email weekly. These precautions will be made to maintain the participant in the study and increase the number of complete follow-ups.

### Intervention

The COPD Web consists of several sections of which one is targeting people with COPD, shown in figure 1. The section targeting people with COPD aims to support self-management and includes, in addition to texts, pictures and films, also interactive components, for example, registration of PA with person-tailored, automatised feedback. Automatised feedback in combination with step counting has been found useful to increase PA in people with COPD.[29] On the website, people with COPD can gain know-how about, for example, PA, physical training, breathing techniques, exacerbation symptoms, advice on when to contact healthcare and how to make everyday activities less strenuous. The content refers to and aligns with the guidelines for COPD care developed and published by the National Board of Health and Welfare in Sweden.[13]

### The intervention group

Participants randomised to the intervention group will be introduced to the COPD Web by a letter containing written information, the password to get access to the website and information on how to create an account. To secure standardised instructions, there will be an instruction movie available on the website (box 1).

The COPD Web will be self-managed. To reduce user problems, one of the researchers (TS) will contact each participant in the first week of intervention. To test the participants' interest for and acceptability of the function of registering PA (steps) on the website, the participants

---

**Box 1  The content of the movie, presenting the administration of the chronic obstructive pulmonary disease Web**

1. Introduction of the website structure, the content in the main headings and functions of the website, for example, how to enlarge or shrink the text, listen to the text and bookmark information of particular interest.
2. Introduction to the section 'Physical activity' (PA). Information about the importance of PA and demonstration of the page for registration of PA (steps) with automated feedback.
3. Information on how to set an initial weekly step goal and instructions to insert the weekly step-count onto the page for registration of PA at the end of each week.

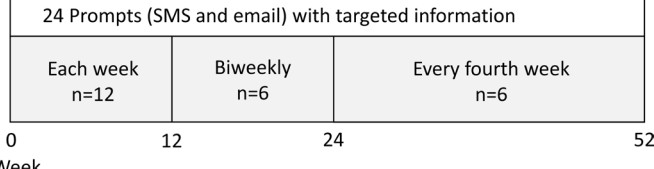

**Figure 2** Distribution of prompts (SMS and email) to participants in the intervention group.

will receive a pedometer with instructions on how it is used.

Throughout the intervention, participants will receive prompts via email and SMS (figure 2). The prompts will include targeted information, referral links to the COPD Web and a reminder to register counted steps to improve adherence to the intervention. Prompts has shown enhanced effectiveness on limited contact interventions targeting health behaviours including PA[30] and proved to be useful also on people with COPD[29] though there is no consensus regarding the number and frequency of prompts. Frequently delivered prompts have been recommended however too excessive appearance may decrease the desired response.[31] Consequently, the frequency of the prompts will be each week at the beginning of the intervention and decrease to biweekly (week 13 to 24) and every fourth week (week 25 to 52). In total, we will deliver 24 different prompts with predetermined content and order to each participant.

### The control group
The control group will, similar to the intervention group, receive a pedometer with instructions, as well as a leaflet about the importance of PA in addition to usual care. In Sweden, the majority of all people with COPD are treated within PHC.[13 15] Usual care within PHC are recommended to include, but are not restricted to, use of long-acting anticholinergics and long-acting β2-agonists with 24 hours duration and support for; smoking cessation, PA and exercise, self-management and nutrition.[13] All participants are permitted to start COPD rehabilitation or other interventions if offered at their PHC unit.

### Outcomes and evaluation
Various methods for data collection including questionnaires, accelerometer, data from medical records (participant's latest pulmonary function test), qualitative interviews and user data from the COPD Web will be used. Box 2 provides an overview of methods for data collection in this study.

### Primary outcome measures
The primary outcome of the effect of the COPD Web is the difference in the level of PA between intervention and control groups at follow-ups (3 and 12 months). Level of PA will be objectively measured seven consecutive days using an accelerometer (DynaPort, McRoberts BV, the Netherlands) and subjectively measured with indicator questions on PA from the National Board of Health and

---

**Box 2    Methods for data collection**

**Physical objectively measured physical activity (PA) level**
► Accelerometer (DynaPort, McRoberts BV (DynaPort, McRoberts BV, The Netherlands) placed on the lower back 24 hours a dayover 7 consecutive days.[34 35]
  – The quantity of PA will be assessed using the mean number of steps per day and the number of days per week that the participant could be considered physically active. physically active is operationally defined as ≥5000 steps per day.
  – The Dynaport accelerometer has been found valid and reliable when used in people with COPD.[34 35]

**Physical subjectively assessed PA level**
► Questionnaire from the National Board of Health And Welfare.[33]
  – The time spent in physical activities such as taking a walk or working in the garden during last week is rated by choosing between prespecified options (no time at all/30–60 min/60–90 min/9–120 min/>120 min).
  – The time spent in physical exercises such as running or doing exercise to keep fit during last week is rated by choosing between prespecified options (no time at all/30–60 min/60–90 min/9–120min/>120 min).
  – The categorical mode of the scale has shown low-to-moderate associations with objectively measured PA level, maximal oxygen uptake, physical performance, balance, cardiovascular biomarkers and self-rated health.[32]

**Health-related quality of life (HRQoL)**
► CRQ-SA The Swedish version of the Self-Administrated Chronic Respiratory Questionnaire.[37]
  – CRQ-SA aims to measure HRQoL in people with chronic respiratory distress. The questionnaire consists of 20 questions divided into four areas (dyspnoea, fatigue, emotional function and control) that are rated on a 7-graded Likert scale. The questions include, for example, 'How often in the last two weeks have you known that you had complete control over your breathing problems?' and 'In the last two weeks, how often have you known that you had low energy?'.[37]
  – CRQ-SA has shown strong responsiveness to changes in HRQoL for people with COPD.[45]

**COPD-related symptoms**
► The questionnaire COPD Assessment Test (CAT).[38]
  – The severity of eight COPD-related symptoms (coughing, the presence of phlegm, feeling of tightness in the chest, breathlessness when walking, limitation in activities, confidence in leaving home, sleep and energy) is rated on a six-grade scale.
  – Evaluated for internal consistency, stability overtime in stable patients and ability to discriminate between stable and exacerbation patients with excellent or very good results.[38]

**Dyspnoea**
► The questionnaire modified Medical Research Council Dyspnea Scale (mMRC).[36]
  – Perceived dyspnoea is rated on a 5-graded Likert scale ranging from 0 ('I just get out of breath when I exert myself greatly' to 4 ('I get out of breath when I wash or get dressed').
  – Evaluated for categorising people with COPD in terms of disability with good results.[46]

**Health economics**
► Self-reported healthcare contacts related to COPD.

Continued

---

## Box 2 Continued

- ► The questionnaire EuroQol fivedimensions questionnaire (EQ-5D).[39]
  - Health status is rated on five items; three items relate to problems in mobility, self-care and usual activities and two items cover the presence and severity of pain and anxiety/depression. Each item is rated on a three-grade scale corresponding to no problem/some or moderate problems/extreme problems.
  - General health is rated on a scale ranging from 0 (worst imaginable health state) to 100 (best possible health state).
  - Evaluation of health economy will be done using EQ-5D to estimate quality-adjusted life (QALY) gained.[40] Also, the number of COPD-related health contacts and hospitalisation that occurs during the intervention will be followed and cost estimated.
  - EQ-5D can discriminate between groups of people with different severity of COPD.[47]

### Implementation

- ► Implementation of the COPD Web.
  - Semistructured interviews will be performed according to a pre-specified interview guide, and user statistics from the website will be analysed.
- ► Fidelity to the intervention.
  - Semistructured interviews will be performed according to a pre-specified interview guide.
- ► Reach.
  - Study-specific documentation including the number of participants who decline to take part in the intervention will be analysed. When appropriate, the reasons to decline will be noted.
- ► Enablers and barriers for the use of web-based support like the COPD Web.
  Semistructured interviews will be performed according to a pre-specified interview guide and analysed.
- ► COPD, chronic obstructive pulmonary disease.

Welfare in Sweden.[32 33] Weekends and weekdays with less than 8 hours of wearing time of the accelerometer and measurements with less than four valid days of measurements will be excluded.[34] The Dynaport accelerometer has been found valid and reliable when used in people with COPD.[34 35]

### Secondary outcome measures

The secondary outcomes of the effect of the COPD Web are the differences between the intervention and control groups at the follow-ups at 3 and 12 months regarding participants' dyspnoea; modified Medical Research Council dyspnoea scale (mMRC),[36] HRQoL; Chronic Respiratory Questionnaire, self-administered (CRQ-SA),[37] and COPD-related symptoms; COPD Assessment Test (CAT).[38] Evaluation of health economics will be done using EQ-5D[39] to estimate quality-adjusted life (QALY) gained, commonly used in economic evaluation.[40] In addition, the number of participant self-reported COPD-related healthcare contacts will be evaluated where a reduction in health consumption indicates a reduced economic burden. Secondary outcomes were chosen according to results in the pilot study and since they cover specific aspects of the content of the COPD

Web. Most of them have previously been used in COPD and a Swedish context.

### User experience and implementation evaluation

For user experience evaluation, data will be collected after 3 months using semistructured individual interviews in a subgroup of participants randomised to intervention. The participants will be asked to take part in an interview at 3 months follow-up. The interviews will include questions regarding unexpected events or consequences of receiving the COPD Web, their use of the website and how this use has influenced their PA behaviour. Study-specific documentation and automatised data on the participants' use of the COPD Web will be collected automatically from the website, for example, the number of visits, pages used and time spent on the website. This will add valuable information to the experience valuation and also make it possible to evaluate the fidelity to the intervention. In order to evaluate the implementation and reach, study-specific documentation including the number of participants who decline to take part in the intervention as well as dropouts will be noted. In addition, the reasons to decline will be noted when appropriate. All participants will also answer study-specific questions regarding other ongoing or started interventions, hospitalisations or exacerbations that could affect the results.

### Data collection, management and analysis

#### Sample size calculation

The sample size was calculated with the premises that a total of 144 participants with COPD would be required to detect a mean difference of 1131 steps with a SD of 2193 steps,[41] α=0.05, β=0.20 (80% power) and a two-tailed test of significance including an estimated dropout rate of 20%.[29] Approximately 10–15 participants will be recruited to individual interviews to have various experiences represented. A wide distribution of age, disease severity and an equal number of women and men will be strived for.

### Randomisation and masking

A permuted block design with a random block size varying from 4 to 8 in a 1:1 allocation ratio will be computer generated to randomise participants. This approach is chosen to achieve balanced and evenly distributed samples. A third party, not involved in data collection or analysis of the results, will perform the randomisation and the result will be stored in sealed envelopes. Thus, the randomisation will be revealed for the researcher when the baseline registration and written informed consent are fulfilled, and the sealed envelope next in order is opened. The researcher then will send a letter containing the result of group allocation, a pedometer, a pamphlet about PA and information about when the participant will be contacted again. The members of the intervention group will, in addition, receive the material and information on how to start using the COPD Web. Due to the character of the intervention, blinding of trial participants will not

be applicable. Furthermore, as all data are self-reported, neither is blinding of outcome assessors applicable.

## Data management and monitoring

To ensure confidentiality, participants with COPD will get a unique identification (ID) when included in the study. The code list linking participants and ID number will be kept separate from the data. Data will be analysed by ID only. All records that contain names or other personal identifiers, such as locator forms and informed consent forms, will be stored separately from study records identified by the ID number. The local database will be secured with a password-protected access system. All data will be coded and reported on group level. Thus, it will not be possible to identify specific participants in the trial. We will use two-pass verification to ensure correct data entry. No interim analyses or stopping guidelines are prespecified. Only the researchers will have access to the final trial dataset.

## Statistics and qualitative analysis

The primary analysis will be an intention-to-treat analysis (including all participants randomised). In addition, a complete case population (participants with full outcome measurements independent on adherence to intervention), and a per-protocol analysis (defined as at least one login besides creating an account on the COPD Web or answering that the SMS and email with referral links have been used at least rarely (1–3 times) at the follow-ups) will be performed. Missing data will be imputed in the intention-to-treat analysis using multiple imputation assuming data is missing at random conditional on participants' severity of disease and self-reported history of exacerbations. This is because the severity of disease and history of exacerbations are known risk factors for future exacerbations and may affect adherence to PA interventions.[42]

The difference in the primary outcome between the intervention and control group will be estimated using multilevel mixed-effects models with subjects at level 1 and PHC units at level 2. PHC units and subjects will be modelled as random effects while group (intervention group versus control group), time and group*time interaction as fixed effects. Estimates of effect sizes will be computed using Cohen's d (d=difference in group means/error SD within). Calculated as the difference between predicted means from the final mixed-effects model for a given pair of groups divided by the estimated within-group error SD in the model with the estimated value of $2\sigma_e^2$, where $\sigma_e^2$ is the residual variance. To judge the quality of the model, we will analyse the residuals. No subgroup or adjusted analyses other than the prespecified complete case and per-protocol analysis will be performed.

The individual interviews will be analysed using qualitative content analysis according to the procedures presented by Graneheim.[43] The interviews transcriptions will be read, coded and categorised by one researcher. Two other researchers will also read and code independently for triangulation. Organisation and labelling of categories will be discussed and modified throughout the process.

## Amendments

Any modifications to the protocol that may influence the conduct of the study, the potential benefit of the participant or may affect participant safety, including changes of study objectives, study design, population, sample sizes, study procedures or significant administrative aspects will require a formal amendment to the protocol. Such modifications will be agreed on by the research group with the final decision by the principal investigator, and if needed to be approved by the local ethics committee.

Administrative changes of the protocol (eg, minor corrections and clarifications) that do not influence how the study is conducted will be agreed on by the research group with the final decision by the principal investigator and will be documented and presented on publication.

## Ethics approval and consent to participate

Ethical approval has been received from the Regional Ethical Review Board in Umeå, Sweden. Dnr 2018-274-31. All participants will receive brief, comprehensible oral and written information, by the Helsinki Declaration.[44] A first informed consent confirms that contact information and latest pulmonary function test from the potential participant can be collected by healthcare professionals and sent to the researchers. The participant will, together with the baseline assessment, send a second and final informed consent to the researcher. The informed consent from operational managers will be sent and stored at the Regional Ethical Review Board in Umeå, Sweden.

## Dissemination

The results of this study will be submitted for publication in peer-reviewed journals and presented at conferences both nationally and internationally as well as to included healthcare professionals, participants and patient organisations for people with COPD.

## Trial registration

Registration of the clinical trial before the enrolment of the first participant was performed. Date of trial initial release 15-11-2018 and published 20-12-2018. ClinicalTrials.gov identifier: NCT03746873. The recruitment began 15-11-2018 and will continue until sufficient power is reached.

## DISCUSSION

This study protocol presents a pragmatic randomised controlled trial with preassessments and postassessments aimed at evaluating the effect of the use of the COPD Web for people with COPD in a PHC context. The study also intends to evaluate the implementation and to identify enablers and barriers to use of web-based support to change behaviour among people with COPD. Currently, despite its proven effectiveness, access to

self-management interventions is limited[2 14] and alternative ways of promoting self-management for people with COPD are warranted. A recent pilot trial has shown that giving people with COPD access to the COPD Web may be an effective short-term strategy to promote self-management that increase levels of PA, promote conceptual knowledge and alter disease management strategies.[24] However, these results need to be confirmed in a definitive large-scale randomised trial, including both short-term and long-term evaluation.

This proposed trial will provide new knowledge to this research area by evaluating the effect of the use of web-based support for increasing access to self-management strategies for people with COPD and determine its effect on clinically relevant outcomes. This trial will include short-term (3 months) and long-term perspectives (12 months) with objectively measured PA in addition to the self-reported PA that will contribute with more knowledge regarding the effect of having access to the COPD Web. PA is of utmost importance, as the level of PA is one of the strongest predictors of mortality among people with COPD.[11 12]

A user experience and implementation evaluation of the intervention will provide novel information and understanding about enablers and barriers for the use of web-based support to change behaviour. This information will increase knowledge of how the process of receiving the intervention can be interpreted. It will also help us draw better conclusions regarding acceptance, fidelity and implementation of the COPD Web.

Guided by the pilot study, prompts will be used to encourage the use of the website during the intervention period.[24] The reminders will provide information with referral links that will appear in a predefined way. Prompts have been proven effective in other setups, but there is no consensus regarding the number of prompts or frequency, especially in a longer perspective.[31] The effect of the prompts will be qualitatively evaluated through the semistructured interviews. The evaluation will answer how the prompts were perceived and if they induced more frequent use and/or changed behaviour regarding PA among the participants. The use of the COPD Web will be automatically registered through the whole intervention since the participants need to log in to access the website. That measure makes it possible to analyse the fidelity to the intervention and answer if there is an association between the use of the COPD Web, for example, time and number of visits and any possible effect.

As the study is designed as a pragmatic trial,[25] the intervention will be self-managed and distance-based to maximise the clinical applicability of the findings. One concern is that there might be participants who do not manage the instructions to create their account and learn how to use the website. However, they will be contacted at the beginning of the intervention to reduce user problems. The pragmatic approach also means that there is no selection on the number, size or location of the recruiting PHC units. Also, the inclusion criteria are set wide with a minimised selection beyond diagnosed COPD that could enhance the recruitment rates and finally increase the clinical applicability of the findings within PHC. One limitation is that the sample size, calculated on PA, will be large enough for evaluation of the PA but may not be powered enough for all secondary outcome or subgroup analyses, the latter much depending on the severity of symptoms among the participants.

In conclusion, this pragmatic randomised trial will provide clinically relevant information on the effect of the use of the COPD Web in people with COPD in a PHC context regarding level of PA, dyspnoea, HRQoL, COPD-related symptoms and health economics in relation to healthcare use, as well as barriers and enablers for using web-based support with solutions such as the COPD Web.

**Contributors** TS has made a direct and substantial contribution to this work by contributing to the conception and design of the study, designing and writing of the protocol. AN has made a direct and substantial contribution to this work by contributing to the conception and design of the study, sample size calculation and choice of statistics, designing and writing of the protocol. SL has made a direct and substantial contribution to this work in providing critical revisions that are important for the intellectual content of the protocol. KW is the principal investigator and has made a direct and substantial contribution to this work by providing the project idea, contributing to the conception and design of the study and by providing critical revisions that are important for the intellectual content of the protocol. All authors have approved the final version of the protocol.

**Funding** This work was supported by The Swedish Research Council, grant number 521-2013-3503 and the Strategic Research Area—Care Science, Umeå University, Sweden, no grant number available.

**Competing interests** None declared.

**Patient consent for publication** Not required.

**Ethics approval** Regional Ethical Review Board in Umeå, Sweden.

**Provenance and peer review** Not commissioned; externally peer reviewed.

**ORCID iD**
Tobias Stenlund http://orcid.org/0000-0003-0569-9490

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
