## [Reviewer comments · BMJ Open]

ARTICLE DETAILS

TITLE (PROVISIONAL)	Web-based support for self-management strategies versus usual care for people with COPD in primary healthcare: a protocol for a randomised, 12 months, parallel-group pragmatic trial.
AUTHORS	Stenlund, Tobias; Nyberg, André; Lundell, Sara; Wadell, Karin

VERSION 1 - REVIEW

REVIEWER	Stephanie Robinson Edith Nourse Rogers Memorial Veterans Hospital
REVIEW RETURNED	28-May-2019

GENERAL COMMENTS	This protocol manuscript describes a pragmatic randomized trial of a web-based self-management intervention for patients with COPD in Sweden. Although there are definite strengths to this manuscript, including its pragmatic nature and inclusion of implementation outcomes, these are unfortunately overshadowed by some very prominent weaknesses that must be addressed before consideration of accepting. 1. One of the biggest issues I struggled with when reading this paper was the rationale for the study. The authors state, rather vaguely, that similar previous trials have been “contradictory” and therefore this study needs to be done. However, the authors have not at all stated any details about these contradictory studies. What were the differences in them? What contradictory results were there? How will this protocol/study help to reconcile these contradictions? Without this information, and regardless of its success in improving self-management and/or PA, this study promises to add to the plethora of contradictory results, rather than help to move the field forward.2. I also note that there is plenty of literature missing from the authors’ review of the literature. Some notable works by researchers such as Moy, Garcia-Aymerich, and Troosters would help to flesh out the scant literature review and support the rationale for this study.3. In its current form, the article almost appears to be written by two distinct writers. There are inconsistencies in abbreviations and notation (e.g., PHC vs. PHCC, 3-month visit vs. t2, eHealth) that should be resolved.4. I applaud the researchers for undertaking a pragmatic trial and wholly believe in the fruitfulness of this type of design. That being said, I am curious if any information will be collected from the control group regarding whether or not they did participate in any other intervention or pulmonary rehabilitation. I think this could be
---

	helpful when trying to interpret findings and potential subgroups of those that did or did not respond well to the intervention. 5. I am curious as to why no measure of self-efficacy is included as a secondary outcome, especially as self-efficacy was cited as an “essential part of COPD management”. This should be clarified and added to the protocol. In a similar vein, all references for the measures and their specific names should be included in the text on page 10. 6. Table 2 details information that will be collected on implementation. However, there is extremely limited, if any, mention of these in the paper. I refer the authors to Curran et al. 2011 on hybrid implementation designs as I believe one of the strengths of this protocol would be to highlight that this is essentially a hybrid effectiveness-implementation type 1 design. Much more detail is necessary when specifying these implementation features, but I strongly believe the effort to detail what information will be collected to assess fidelity, reach, and facilitators and barriers will be a real selling point of this manuscript and study. 7. Why was the study powered on 1131 steps? This should be explained. 8. The authors should clarify they plan to conduct mixed multilevel models. 9. Finally, but very importantly, this article was difficult to read. There were many typos, grammatical errors, and writing faux-pas. For example, paragraphs should include at least 3 sentences in them. I would strongly recommend a heavy edit to increase the readability. Additionally, the authors may want to consider renaming the intervention to “COPD Web” as opposed to “the COPD Web” to further increase readability.
--	---

REVIEWER	Ian Yang The Prince Charles Hospital and The University of Queensland
REVIEW RETURNED	14-Jun-2019

GENERAL COMMENTS	Thank you for inviting me to review this interesting paper. The role of web-based platforms for COPD self-management and care are not yet defined in clinical trials. This pragmatic RCT will address the benefits of use of a web platform for delivering self-management advice for patients with COPD. The trial is well-designed, with clear methodologies and outcomes planned. Major comments: Nil Minor comments: Use of the web platform: How often will patients be expected to access the website? If there is a period of inactivity (e.g. prolonged lack of use of the website or other devices), then will this be signaled to the investigators so that they can respond and check in with the patient about adherence to the intervention?
--

VERSION 1 – AUTHOR RESPONSE

Respond to the reviewers.

We thank the reviewers for all the relevant and valuable comments and suggestions. Below are the listed comments from the reviewers with our response and the changes made in the revised manuscript highlighted. The revised protocol is, thanks to the relevant comments, improved.

Reviewer 1.

1. One of the biggest issues I struggled with when reading this paper was the rationale for the study. The authors state, rather vaguely, that similar previous trials have been “contradictory” and therefore this study needs to be done. However, the authors have not at all stated any details about these contradictory studies. What were the differences in them? What contradictory results were there? How will this protocol/study help to reconcile these contradictions? Without this information, and regardless of its success in improving self-management and/or PA, this study promises to add to the plethora of contradictory results, rather than help to move the field forward.

Answer: Due to the aforementioned issue we have tried to clarify the rationale in the introduction. We have accordingly included additional references. Voncken-Brewster et al. (2015) showed no results for subjectively answered PA after 6 months of web-based, computer-tailored self-management intervention. One explanation could be the participant’s low exposure to the application. Moy et al. (2015) and Wan et al. (2017) on the other hand showed improved objectively measured PA when a web-based solution intervention and pedometers were used. A review McCabe et al. (2017) states that interventions aimed at facilitating self-management in people with COPD and delivered via smart technology improved levels of activity up to six months but that no firm conclusions can be drawn and that the improvement may not be sustained over a long duration. The long-term evaluation is therefore one of the most important contributions from this protocol that hopefully can help to move this field forward.

Adjustments made in the introduction

“However, studies evaluating whether a web-based solution as the COPD Web could be used to promote self-management strategies to support increased PA in people with COPD are contradictory. One showed no effect on PA₁₉ while other studies showed improved PA₂₀₋₂₂ but that the improvement may not be sustained over a long duration.²²”

“To know whether this is true also for a larger COPD population an adequately powered randomised controlled trial with short and long-term evaluation is needed.”

2. I also note that there is plenty of literature missing from the authors' review of the literature. Some notable works by researchers such as Moy, Garcia-Aymerich, and Troosters would help to flesh out the scant literature review and support the rationale for this study.

Answer. We thank for the suggestion and have added some valuable references to the introduction.

Added references in the manuscript:

"9. Garcia-Aymerich J, Pitta F. Promoting Regular Physical Activity in Pulmonary Rehabilitation. *Clinics in Chest Medicine* 2014;35(2):363-68. doi: 10.1016/j.ccm.2014.02.011"

"10. Troosters T, Sciruba F, Battaglia S, et al. Physical inactivity in patients with COPD, a controlled multi-center pilot-study. *Respir Med* 2010;104(7):1005-11. doi: 10.1016/j.rmed.2010.01.012 [published Online First: 2010/02/20]"

"17. Loeckx M, Rabinovich RA, Demeyer H, et al. Smartphone-Based Physical Activity Telecoaching in Chronic Obstructive Pulmonary Disease: Mixed-Methods Study on Patient Experiences and Lessons for Implementation. *JMIR mHealth and uHealth* 2018;6(12):e200. doi: 10.2196/mhealth.9774"

"19. Voncken-Brewster V, Tange H, de Vries H, et al. A randomized controlled trial evaluating the effectiveness of a web-based, computer-tailored self-management intervention for people with or at risk for COPD. *International journal of chronic obstructive pulmonary disease* 2015;10:1061-73. doi: 10.2147/copd.s81295 [published Online First: 2015/06/20]"

"20. Moy ML, Collins RJ, Martinez CH, et al. An Internet-Mediated Pedometer-Based Program Improves Health-Related Quality-of-Life Domains and Daily Step Counts in COPD: A Randomized Controlled Trial. *Chest* 2015;148(1):128-37. doi: 10.1378/chest.14-1466 [published Online First: 2015/03/27]"

"21. Wan ES, Kantorowski A, Homsy D, et al. Promoting physical activity in COPD: Insights from a randomized trial of a web-based intervention and pedometer use. *Respir Med* 2017;130:102-10. doi: 10.1016/j.rmed.2017.07.057 [published Online First: 2017/12/06]"

"22. McCabe C, McCann M, Brady AM. Computer and mobile technology interventions for self-management in chronic obstructive pulmonary disease. *Cochrane Database of Systematic Reviews* 2017(5) doi: 10.1002/14651858.CD011425.pub2"

3. In its current form, the article almost appears to be written by two distinct writers. There are inconsistencies in abbreviations and notation (e.g., PHC vs. PHCC, 3-month visit vs. t2, eHealth) that should be resolved.

Answer. PHC was an abbreviation for Primary healthcare and PHCC an abbreviation for primary healthcare center. Since this seems to be difficult to interpret, we now only use one abbreviation. We have changed PHCC to PHC unit or just unit throughout the manuscript. The other inconsistencies mentioned are now resolved.

4. I applaud the researchers for undertaking a pragmatic trial and wholly believe in the fruitfulness of this type of design. That being said, I am curious if any information will be collected from the control group regarding whether or not they did participate in any other intervention or pulmonary rehabilitation. I think this could be helpful when trying to interpret findings and potential subgroups of those that did or did not respond well to the intervention.

Answer. Thanks for the appreciation. We will collect information regarding if the participants has any ongoing, started or ended intervention of any kind besides the one given in the study at baseline, 3 and 12 month. We will also collect information if there have been any exacerbations or COPD related events for which they sought medical help.

Adjustment made in the manuscript in section Implementation and user experience evaluation

“All participants will also with the questionnaires answer study-specific questions regarding other ongoing or started interventions, hospitalisations or exacerbations that could affect the results.”

5. I am curious as to why no measure of self-efficacy is included as a secondary outcome, especially as self-efficacy was cited as an “essential part of COPD management”. This should be clarified and added to the protocol. In a similar vein, all references for the measures and their specific names should be included in the text on page 10.

Answer. We understand the legitimate question due to a mistake on our part. Unfortunately, we mentioned self-efficacy that was a part in the pilot study. Self-efficacy to perform PA was assessed in

the pilot study using the ESES: Exercise Self-Efficacy Scale. Self-efficacy showed no results in the pilot study and the intention was therefore not to focus on self-efficacy in this study. We have now made a change in the introduction.

Adjustment made in the manuscript in section introduction

“Self-management strategies, including strategies to promote change in health behaviour by increasing the individual’s knowledge and skills and their confidence in successfully managing their disease, is therefore now an essential part of COPD management.⁵”

6. Table 2 details information that will be collected on implementation. However, there is extremely limited, if any, mention of these in the paper. I refer the authors to Curran et al. 2011 on hybrid implementation designs as I believe one of the strengths of this protocol would be to highlight that this is essentially a hybrid effectiveness-implementation type 1 design. Much more detail is necessary when specifying these implementation features, but I strongly believe the effort to detail what information will be collected to assess fidelity, reach, and facilitators and barriers will be a real selling point of this manuscript and study.

Answer

We would like to thank the reviewer for the information regarding hybrid effectiveness-implementation design and the Curran et al. reference. Due to the good suggestion, we have extended the information regarding the implementation in outcomes and made some adjustments in the manuscript. Fidelity to the intervention will be collected through study-specific documentation and automatised collected data on the participants’ use of the COPD Web, e.g., number of visits, which part of the website that was used and time spent on the site. In order to evaluate the implementation and who is reached and not reached study-specific documentation including the number of participants who decline to take part in the intervention will be noted by the healthcare personell and the researchers. Also the reasons to decline or drop out will be noted when this is appropriate and possible.

Despite some similaritys we doubt that we fully complies with the criteria for a effectiveness-implementation design e.g. we don’t include a full implementation process evaluation. We have therefore not stated that this protocol is according a effectiveness-implemtation design. However, we thank the reviewer for the new information that we will strive to use in forthcoming studies.

Adjustment made in the manuscript in section Trial design

“The design is a pragmatic randomised controlled trial with pre- and post-assessments (3 months and 12 months) and with a user experience and implementation evaluation. The user experience and implementation evaluation is a necessary complement that will be performed to understand more about enablers and barriers for behavior change using web-based solutions like the COPD Web.”

Adjustment made in the manuscript in section outcomes

“This will add valuable information to the experience valuation but also make it possible to evaluate the fidelity to the intervention. In order to evaluate the implementation and who is reached and not reached study-specific documentation including the number of participants who decline to take part in the intervention or drop outs will be noted. In addition, when appropriate, will the reasons to decline also be noted. All participants will also with the questionnaires answer study-specific questions regarding other ongoing or started interventions, hospitalisations or exacerbations that could affect the results.”

7. Why was the study powered on 1131 steps? This should be explained.

Answer

In the pilot study, PA was collected through questionnaires at 3 and 12 months but not objectively measured. We therefore conducted a literature search and found a study by Demeyer et al. (2016) who reported the data required for the power calculation. Demeyer et al. (2016) reported a mean difference of 1131 steps in patients with COPD in their study regarding minimal important differences. Since we have already referred to Demeyer et al. in the manuscript, we believe that further explanation is not necessary. However, we have moved the reference number to better connect the mean differences and standard deviation to the aforementioned study.

Adjustments made in the manuscript section sample size calculation

“The sample size was calculated with the premises that a total of 144 participants with COPD would be required to detect a mean difference of 1131 steps with a standard deviation of 2193 steps⁴⁶, $\alpha = 0.05$, $\beta = 0.20$ (80% power), and a two-tailed test of significance including an estimated dropout rate of 20%.³¹”

8. The authors should clarify they plan to conduct mixed multilevel models.

Answer

We have consulted a statistician who, based on his expertise, has provided a more thoroughly description of the model in the section for statistics. We hope this addition will clarify the ambiguity that existed in the previous manuscript.

Adjustment made in the manuscript in section outcomes

“The difference in primary outcome between intervention and control group will be estimated using multilevel mixed effects models with subjects at level 1 and PHC units at level 2. PHC units and subjects will be modelled as random effects while group (intervention group vs control group), time and group*time interaction as fixed effects.”

9. Finally, but very importantly, this article was difficult to read. There were many typos, grammatical errors, and writing faux-pas. For example, paragraphs should include at least 3 sentences in them. I would strongly recommend a heavy edit to increase the readability. Additionally, the authors may want to consider renaming the intervention to “COPD Web” as opposed to “the COPD Web” to further increase readability.

Answer

We have carefully reviewed the script to correct typos, grammatical errors, and made changes to increase readability. We have considered renaming the intervention to “COPD Web” but we still believe that “the COPD Web” is more correct in most occasions and that is therefore our choice. We have, however, reduced the use of “the COPD Web” since it sometimes may has been perceived as redundant information.

Reviewer 2

Use of the web platform: How often will patients be expected to access the website? If there is a period of inactivity (e.g. prolonged lack of use of the website or other devices), then will this be

signaled to the investigators so that they can respond and check in with the patient about adherence to the intervention?

Answer

We have no expectations regarding the participant's use of the website in terms of time or number of events. We have in the statistics section regarding adherence to the intervention described that "a per-protocol analysis (defined as at least one login besides creating an account on the COPD Web or answering that the SMS and email with referral links have been used at least rarely (1-3 times) at the follow-ups) will be performed." During the study, there will not be any warnings to the investigator due to participant's inactivity on the COPD Web. Such warnings could be valuable and something to implement in a future clinical setting. However, one issue to solve for a future warning system is that the SMS and email with referral links are constructed in a way so that the participant can get a quick glance of a specific page on the website without having to log in. This technical solution with referral links was chosen to simplify the access of information for less experienced users. At present, a participant that only uses the referral links in the messages but don't click on or use anything else on the website will appear as inactive. The adherence to the intervention is therefore based on the automatic data from the website including time and number of login plus specific questions at 3 and 12 months follow-up regarding the number of SMS and e-mail with referral links used.

We have not made any adjustments in the manuscript.